# Effect of Plasma Argon Pretreatment on the Surface Properties of AZ31 Magnesium Alloy

**DOI:** 10.3390/ma16062327

**Published:** 2023-03-14

**Authors:** Cecilia Montero, Cristián Gino Ramírez, Lisa Muñoz, Mamié Sancy, Manuel Azócar, Marcos Flores, Alfredo Artigas, José H. Zagal, Xiaorong Zhou, Alberto Monsalve, Maritza Páez

**Affiliations:** 1Departamento de Ingeniería Metalúrgica, Facultad de Ingeniería, Universidad de Santiago de Chile, Alameda 3363, Santiago 9170022, Chile; 2Facultad de Ciencias, Instituto de Química, Pontificia Universidad Católica de Valparaíso, Valparaíso 2373223, Chile; 3Escuela de Construcción Civil, Pontificia Universidad Católica de Chile, Vicuña Mackenna 4860, Santiago 7820436, Chile; 4Departamento de Química de Materiales, Facultad de Química y Biología, Universidad de Santiago de Chile, Alameda 3363, Estación Central, Santiago 9170022, Chile; 5Departamento de Física, Facultad de Ciencias Físicas y Matemáticas, Universidad de Chile, Beauchef 850, Santiago 8370415, Chile; 6Corrosion and Protection Centre, School of Materials, The University of Manchester, Manchester M13 9PL, UK

**Keywords:** magnesium alloy, AZ31 alloy, argon plasma, surface treatment, corrosion

## Abstract

Climate change has evidenced the need to reduce carbon dioxide emissions into the atmosphere, and so for transport applications, lighter weight alloys have been studied, such as magnesium alloys. However, they are susceptible to corrosion; therefore, surface treatments have been extensively studied. In this work, the influence of argon plasma pretreatment on the surface properties of an AZ31 magnesium alloy focus on the enhancement of the reactivity of the surface, which was examined by surface analysis techniques, electrochemical techniques, and gravimetric measurements. The samples were polished and exposed to argon plasma for two minutes in order to activate the surface. Contact angle measurements revealed higher surface energy after applying the pretreatment, and atomic force microscopy showed a roughness increase, while X-Ray photoelectron spectroscopy showed a chemical change on the surface, where after pretreatment the oxygen species increased. Electrochemical measurements showed that surface pretreatment does not affect the corrosion mechanism of the alloy, while electrochemical impedance spectroscopy reveals an increase in the original thickness of the surface film. This increase is likely associated with the high reactivity that the plasma pretreatment confers to the surface of the AZ31 alloy, affecting the extent of oxide formation and, consequently, the increase in its protection capacity. The weight loss measurements support the effect of the plasma pretreatment on the oxide thickness since the corrosion rate of the pretreated AZ31 specimens was lower than that of those that did not receive the surface pretreatment.

## 1. Introduction

Magnesium alloys are the lightest engineering materials, with the best strength-to-weight ratio [1], but their poor corrosion resistance makes their use limited in the transportation industries [2]. Thus, surface treatments on magnesium alloys have been extensively studied [1,3], because they can improve anticorrosion as well as mechanical properties [4]. Surface treatments that use chromates, such as conversion coatings, are widely used for anti-corrosive treatments for light alloys like magnesium. However, the carcinogenic nature of hexavalent chromium makes developing chromium-free treatments an urgent matter, especially when such a restriction is included in new regulations [1]. Phosphate-based conversion treatments with or without metallic compounds (Co, Ni, Mn, Ca, and Zn) are the most popular, with many patents and research into them in recent years [1]. Some of these Cr(VI)-free treatments aim to inhibit active corrosion (self-healing), which is particularly useful when these are mechanically damaged. An example is the conversion treatment with phosphate and permanganate, which showed an equivalent or slightly higher passivating capacity than the Cr(VI)-based treatments for magnesium alloys but presented lower protection on pure Mg [5].

Ssurface treatments based on plasma have recently attracted the attention of several researchers because they are simpler, eco-friendly, and they can modify the surface, removing organic contaminants, improve the surface wettability [6,7], and improve the interaction between the metal and organic species [7]. Plasma can also favor the generation of free radicals on alloy surfaces [6]. The increase in surface energy has been observed on different alloys, attributed to the hydroxylation of the surface components [3]. Different gases or their mixture for plasma pretreatment modify the surface. The physical or chemical change depends on the type of gas used. Generally, inert gases, such as helium, argon, and krypton, or reducing gases, such as hydrogen, or reactive gases, such as oxygen, are used for surface modification [8,9,10]. Inert gases are used to form free radicals on the surface leaving active sites for a later reaction. On the other hand, if the plasma is an inert gas such as argon or helium, the surface can contain many stable radicals that can persist even after exposure to a reactive gas [7]. It has been shown that pretreatment with an oxygen-argon gas mixture promotes complete oxidation of the AZ91 surface resulting in a nanoporous surface [3], unlike the effect of pretreatment on other alloys, such as stainless steel, where surface roughness decreases [6,8,11]. Tang et al. [11] employed atmospheric plasma to modify the surface of AISI 304L stainless steel, with the result of increasing wettability and surface free energy. Other researchers, like Shin et al. [12], improved the surface adhesion of paint coatings using argon plasma at atmospheric pressure. Moreover, Kim used an atmospheric pressure plasma jet to modify aluminum, copper, and stainless-steel surfaces, using contact angle analysis to determine surface activation properties [10]. Mui et al. [13] improved the adhesion of a polyurethane pain on an aluminum alloy, while Muñoz et al. [7] improved the adhesion of a poly(methyl-methacrylate) on an AA2024 alloy with argon plasma. Xu et al. [14] report an improvement in polyphenylene adhesion, and an augmentation to the fracture resistance of the composite structure. On the other hand, Yoshida et al. [15] used oxygen plasma to improve the adhesion of the polyether ether ketone film to the copper surface, increasing hydrophilicity and surface roughness.

In this study, argon plasma at low pressure is used for the surface modification and increases the reactivity of the AZ31 magnesium alloy. This surface pretreatment could be used for improving the interaction between the surface and a future organic coating as mentioned previously [7,12,13]. Evaluation of the properties of the resultant surfaces was performed using contact angle measurement and atomic force microscopy (AFM). For corrosion study gravimetric and electrochemical measurements were performed.

## 2. Materials and Methods

### 2.1. Metal Samples

AZ31 magnesium alloy (wt.%: 94.94 Mg, 3.50 Al, 1.00 Zn, 0.45 Mn, 0.08 Fe, 0.02 Cu, 0.01 Ni) sheet with a thickness of 1.5 mm was used in the present study.

The samples were cut depending on the experiment, mechanically polished with SiC paper up to 4000 grit, washed with ethanol, sonicated with ethanol, and dried in a cool airstream. All experiments were done at least in triplicate to get reproducible results.

### 2.2. Argon Plasma Pretreatment

In order to activate the surface, argon plasma pretreatment was performed on half of the samples with a Plasma Prep III device (SPI, Plasma Prep III Solid State, Manufacturer, West Chester, PA, USA) that consisted of an RF supply (13.56 MHz) in a dielectric quartz tube in a hollow electrode. The gas flowing out of the nozzle formed a plasma jet of pure argon (5 L/min). The power applied to the electrode was adjusted to 80 W at a pressure of around 150 mTorr (20 Pa). This power was chosen because it is the one that provides enough power to activate the surface without etching it. The exposition time was two minutes, according to the protocol proposed by Muñoz et al. [7].

For easier identification, specimens before plasma pretreatment are called AZ, and specimens after plasma pretreatment are called AZ + Ar-P.

### 2.3. Surface Analysis

#### 2.3.1. Contact Angle

The contact angle was measured using a contact angle device (Drop Shape Analyzer DSA25S, KRUSS, Hamburg, Germany) controlled by ADVANCE software (KRÜSS), using the sessile drop method. As a result, 8 μL drops of deionized water or diiodomethane were deposited on the surface. The surface energy was calculated using the WORK method (Owens, Wendt, Rabel and Kaelble model), which was determined after 2 min of exposition of the sample to the argon plasma jet.

#### 2.3.2. Atomic Force Microscopy

Surface topography was measured using an atomic force microscope (AFM, Scienta Omicron, Uppsala, Sweden) and shown by the root mean square value. AFM was performed with a microscope operated on UHV conditions, 10^−7^ mbar (SPM1 Omicron). The AFM-contact mode was used with PP-CONTR tips (NanosensorsTM, Neuchatel, Switzerland), which has a radius of curvature of 10 nm and an elastic constant of 0.02–0.77 N/m. For surface images, WSxM 5.0 software (Laboratorio de Nuevas Microscopías, Universidad Autónoma de Madrid, Madrid, Spain) was used [16].

#### 2.3.3. X-ray Photoelectron Spectroscopy

A FlexPS SPECS spectrometer (SPECS Surface Nano Analysis GmbH Voltastrasse 5, Berlin, Germany) allowed us to get the XPS analysis, with an energy scale calibrated with the binding energy of C 1s peak of adventitious carbon at 284.8 eV. The fitting and deconvolution of the curve were performed after removing the Shirley-type background. For the high resolution of O 1s peak Gauss fit was used.

### 2.4. Electrochemical Measurements

A three-electrode electrochemical cell was used for all electrochemical measurements. A graphite rod and a mercurous sulfate electrode (MSE) were used as counter and reference electrodes, respectively. The metal samples, previously cut (1.5 cm × 5.0 cm) and polished, according to the protocol presented in sample preparation, were used as working electrodes with an exposure area of 0.76 cm^2^, which was limited by an O-ring. Electrochemical measurements were carried out using a Gamry 600+ reference potentiostat /galvanostat, and 0.1 M of sodium sulfate (Na_2_SO_4_, reagent grade) was used as electrolyte. All experiments were done at room temperature in a Faraday cage, by triplicate. Open circuit potential (E_OC_) was measured against the reference electrode during 240 h, which were recorded every 0.5 s. Potentiodynamic polarization measurements were carried out at a scan rate of 0.1 mV s^−1^ after 240 h, which were performed in two steps, anodic and cathodic scan, from E_OC_. Potentiostatic EIS measurements were carried out from 20 kHz to 3 mHz, with an amplitude of 20 mV, recording eight data points per decade of frequency, which were collected after different immersion times, up to 240 h.

### 2.5. Weight Loss

The mass determinations of the samples were made using an analytical balance (readability 0.01/0.1 mg, Radwag AS 82/220.R2, Radwag, Radom, Poland) to estimate the loss mass until the average mass was constant, as described ASTM G1-03 standard [17]. The dimensions of the samples were also measured with a caliper and the mass was registered between 3 h to 240 h in triplicate.

## 3. Results and Discussion

### 3.1. Surface Characterization

#### 3.1.1. Contact Angle

Table 1 shows the contact angle data (water, θ_w_, and diiodomethane, θ_D_) before and after 120 s of exposition at the argon plasma. The surface energy and its components (γ_LV_; γ^d^_LV,_ dispersive energy; γ^p^_LV_, polar energy) were calculated from the data. Before the surface exposition to argon plasma, the contact angle of the water on the surface was 37.60° ± 8.37°, decreasing to 5.31° ± 0.98° after 120 s of exposure, and in consequence, an increase of the surface energy. This increase in wettability was expected because surface plasma modification favors the formation of functional groups, in this case, hydroxyls and oxides on the surface of the alloy [8]. These results indicate that the hydrophilicity of the surface increases, which is consistent with previous studies [6,8,11].

Mrad et al. [18] reported that increasing surface hydrophilicity increases wettability with polar organic compounds due to increased surface hydroxyl groups. According to the literature, the values of the contact angle of water, diiodomethane, and surface energy for mechanically polished AZ31 alloy are 24.4° ± 2.4°, 33.0° ± 4.4° and 72.65 mN/m ± 1.42 mN/m, respectively [19]. Further, a contact angle close to 32.2° has been determined for pure magnesium [20]. All contact angle values of magnesium and AZ31 alloy are less than 45°, so it can be said that magnesium and its AZ31 alloy are already hydrophilic due to the hydroxide that forms on the surface [21].

For some alloys, such as 316LVM stainless steel and AA2024 aluminum alloy, decreases in water contact angles are reported when surfaces are previously exposed to a plasma pretreatment. Sönmez [6] studied stainless steel with oxygen pretreatment and argon plasma separately. Initially, the contact angle was 50.4° ± 8.3°, which decreased to 3.67° ± 0.79° and 5.90° ± 2.60° after 30 min of exposure to plasma of argon and oxygen, respectively. In addition, for surfaces of AA2024 alloys, Muñoz et al. [7] reported a decrease in the contact angle of water from 56.9° ± 0.1° to 17.7° ± 6.7° after two minutes of exposure to plasma.

#### 3.1.2. Atomic Force Microscopy

The results of the AFM images are shown in Figure 1. Surfaces that had been only polished had an RMS of about 4 ± 1 nm on a 500 nm × 500 nm area, while for surfaces after 120 s of exposition to plasma argon, RMS was 16 ± 5 nm at the same scale; therefore, the roughness increased when the pretreatment was applied.

Lin et al. [8] showed the results for stainless steel with a mixture of gasses at atmospheric pressure (Ar, N_2_, and O_2_) exposed for 180 s, resulting in a lower average roughness but more peaks and valleys, while Sömnez et al. [6] exposed the stainless steel for 30 min, resulting in a surface etching and a decrease in surface roughness. In the case of magnesium alloys, this study is very novel. Tiyyagura et al. [3] studied the combination of oxygen and argon plasma on the surface, a procedure that led to a rougher surface. As mentioned in the results of the contact angle measurement, the surface is enriched with oxides and hydroxyls due to the treatment with argon plasma. This is because plasma-induced etching is selective in removing adsorbed contaminants, giving an exposed metallic outermost surface. Regarding the reorganization of the material on the surface, this selective etching can induce the reorganization of the metallic material on the surface of the sample, resulting in the roughness being increased due to the formation of nanostructures, as seen in Figure 1. Thus, the impact of the argon ions may create defects on the surface and/or give activation energy for surface diffusion, contributing to the shape of the oxide layer [3,22]. These structures are susceptible to suffer surface oxidation or hydroxylation.

#### 3.1.3. X-ray Photoelectron Spectroscopy

Figure 2a,b show the XPS survey of AZ31 alloy before and after the argon pretreatment. The high-resolution O 1s peak of AZ and AZ + Ar-P samples, in Figure 2c,d—which reveals the chemical change on the surface—showed two peaks, one at 531 eV associated with OH^−^ species on the surface [3] (green line) and a second peak at 530 eV that was attributed to the surface oxide [23] (red line). Before the argon pretreatment, the O/OH^−^ ratio was ~1, which increased after the pretreatment. The increase in oxide can be related to a higher surface reactivity due to the argon-plasma treatment, which is more susceptible to oxidation in the presence of air. This phenomenon is consistent with the contact angle measurements and AFM images, as described by Tiyyagura et al. [3], who reported that the magnesium alloy surfaces prior to the argon-plasma pretreatment presented mainly hydroxyl groups and that, after pretreatment, oxygen species increased considerably. These last results agree with the results reported in this work. Figure 2e,f show the Mg 2p high-resolution spectra for AZ and AZ + Ar-P, respectively, which in turn showed the same trend that in O 1 s spectra, where the peak at 49.3 eV associated with Mg(OH)_2_ (green line) [22,23] decreases after the pretreatment with respect to the peak at 49.7 eV associated with MgO (red line) [24,25], confirming the previous result.

### 3.2. Electrochemical Results

#### 3.2.1. Open Circuit Potential

The open circuit potential (OCP) measurements of the AZ and AZ + Ar-P samples were performed after different exposure times to the electrolyte, up to a maximum exposure time of 240 h. The behavior of the OCP over time is presented in Figure 3. For exposure times of less than 48 h, the OCP moves progressively towards more positive values; for over 48 h of exposure to the electrolyte, the OCP reaches a constant value close to −1.86 V vs. SSE. This behavior of the OCP over time agrees with that reported by Leleu et al. [2]. OCP stabilization has been commonly attributed to developing a partially protective film on the metal surface composed of MgO/Mg(OH)_2_ [2,26,27,28]. The behavior of the pH with the exposure time at the surface-electrolyte interface also coincides with that reported by Leleu et al. [2]; the pH increases from 7 to ~10 after 24 h of immersion. The increase in pH may be related to the formation and dissolution of the Mg(OH)_2_ film and, consequently, to the release of OH^-^ to the electrolyte, as described by Baril et al. [29]. According to previous studies, the maximum potential reached corresponds to the stabilization of the hydroxylated species and is related to a pH value of around 10.5 [30]. The OCP behaviors of the AZ and AZ + Ar-P samples are similar, which is related to the magnitude of the impact of the plasma pretreatment on the surface roughness. As revealed by the AFM micrographs in the Figure 2, the changes in roughness promoted by the plasma pretreatment are nanometers.

In this investigation, argon plasma pretreatment was used only to activate the surface, and changes in roughness at the nanometer scale did not affect the OCP values. There needs to be more than a difference in roughness to resolve the small changes that can affect the system. For this reason, other electrochemical techniques are used to determine changes in the electrochemical behavior of samples.

#### 3.2.2. Potentiodynamic Polarization

Figure 4 shows the potentiodynamic polarization curves of AZ and AZ + Ar-P samples after different immersion times, which reveals that the corrosion potential became slightly more positive, from −1.96 ± 0.01 V vs. MSE to −1.82 ± 0.02 V vs. MSE, after 168 h of immersion, as also shown in Table 2. This potential shift can be attributed to the progressive formation of an oxide film that is partially protective and which decreases the active surface area [25,26,29,30,31,32,33]. Figure 4 shows an increase of the corrosion current densities (i_corr_) during the first 24 h with the application of the plasma pretreatment, with no significant difference between the samples between 72–168 h. However, after 240 h, an increase in the i_corr_ of AZ sample is evident.

In the anodic branch, an inflection point in the potential is revealed, usually attributed to the breakdown of the oxide formed on the surface, also known as the potential of the breakpoint (E_B_) [2,34,35]. When argon plasma pretreatment is applied, it is important to note that the E_B_ is shifted to more positive potentials as time progresses, which can be attributed to a more extensive pseudo-passivity range. Conversely, for AZ sample an increase in the pseudo-passivity range was observed in the first 72 h and then a rapid decrease, suggesting that a more stable oxide is formed on the surface after applying the argon plasma pretreatment over a longer time.

Figure 5 shows the anodic current density of AZ + Ar-P samples at E = E_oc_ + 50 mV after 3 h of immersion (i_η_ = 50 mV), revealing that it was higher for AZ + Ar-P than AZ samples, which can be explained by the initial reactivity of the surface. After 24 h the i_η_ = 50 mV was stabilized for both samples, however, after 240 h of immersion, the i_η_ = 50 mV of AZ samples increases, as well as the dispersion of the results. According to Feliu et al. [36], the great dispersion in the samples can be caused by the growth characteristics of the corrosion products, which are extremely sensitive to fortuitous and minimal variations in the film formation process. This could indicate that the pretreatment leaves a more stable surface.

#### 3.2.3. Electrochemical Impedance Spectroscopy

Figure 6 shows Nyquist diagrams for AZ and AZ + Ar-P samples at E = E_OC_, which reveals two capacitive loops, in the high-frequency range (HF) and in the medium frequency range (MF), and an inductive loop in the low-frequency range (LF). It has been reported that the capacitive loop at HF is associated with a charge transfer resistance (R_CT_) and the capacitance of the film on the metal surface [2,30,32,37], which increases with the exposure time. This can be attributed to the decrease in the free surface area due to the increase of the thickness of the oxide film [2]. Baril et al. [29] studied the corrosion mechanism of pure magnesium in sodium sulfate solution, and concluded that the formation of the oxide layer has a double structure: a thin internal layer of MgO in contact with the metal substrate, which has a protective behavior; and an external porous layer of Mg(OH)_2_. The authors attributed the increase in impedance response to an increase in the ratio of the inner MgO film covering the area of the outer film [29]. The second loop at MF range has been related to the diffusion of charged species, such as Mg^2+^ ions or O^2−^, through the porous hydroxide layer [2,32,37]. The inductive loop at LF has been usually attributed to the relaxation process of the intermediates species adsorbed on the surface, which some authors related to Mg^+^ [2,23,29,30,32,37], explained as the time that elapse, after the system disturbance and before the new steady state coverage is established and the corresponding current flows [23].

The impedance responses of the AZ31 samples after different exposure times to the electrolyte depends on the surface pretreatment. For the AZ samples, an increase in impedance was determined after 120 h, which decreased after 168 h, increasing again after 240 h. This behavior reveals a low protective efficiency of the oxide film. In contrast, the AZ + Ar-P samples showed a progressive increase in impedance up to 240 h, which suggests an enhancement of the protective properties associated with the Ar plasma pre-treatment. Ascencio et al. [23] studied the impedance response of the WE43 magnesium alloy, reporting that the increase in impedance was related to the formation of the oxide layer and the decrease in impedance to the rupture of the layer, which can favor the localized pitting on the alloy surface. Furthermore, Xin et al. [38] studied the corrosion behavior of AZ91 magnesium alloy in an SBF solution, and attributed the decrease in impedance to the appearance of pitting corrosion. Leleu et al. [2] proposed that the impedance behavior of the magnesium and magnesium alloys was not easily modeled by resistors, capacitors, and inductances because the diffusion process of Mg^2+^ and the presence of adsorbed intermediates were both complex. From the graphical methods, the authors determined that the film formed on the surface has a double structure [37]. In addition, Leleu et al. [2,31] reported that two effects controlled the corrosion process, the dissolution of the Mg-rich matrix that led to a progressive surface coverage by a protective film that decreased the active surface area, and the galvanic couplings between the β-phase (Mg_17_Al_12_) or intermetallic particles (of the Mn_8_Al_5_ type) and the Mg matrix. Furthermore, they found that incorporating the alloying elements proceeds in the inner oxide film, which is thinner and more stable.

It is possible to obtain information about the properties and formation of the film formed on the surface using the impedance data in the HF through the complex capacitive response:(1)$$C\left(\omega \right)=\frac{1}{j\omega \left[Z\left(\omega \right)-R_{e}\right]}$$
where *ω* = 2πf, *Z* is the imaginary part of the impedance, and *R_e_* is the resistance of the electrolyte, which can be determined by extrapolating the impedance in the HF to the real axes of the Nyquist diagram, as reported in Table 3. Figure 7a shows the complex capacitance, which allows the estimation of the oxide capacitance (*C_ox_*). Table 3 and Figure 7b show the evolution of these values. For AZ samples the value of *C_ox_* varied around 4 μF cm^2^. For AZ + Ar-P samples, the *C_ox_* values showed a higher variation, between 1 to 7 μF cm^2^, which can be attributed to the higher reactivity of the surface due to the argon plasma effect. It has been reported that the *C_o_*_x_ can be associated with the thickness of the oxide, with the relationship:(2)$$C_{ox}=\frac{\varepsilon \varepsilon_{0}}{\delta_{ox}}$$
where ε_0_ is the vacuum permittivity (8.85 × 10^−14^ Fcm^−1^), ε is the dielectric permittivity, which was assumed mainly for the MgO (ε = 9, [37]), and δ_ox_ is the oxide thickness.

Figure 8 shows the evolution of the layer thickness during exposure, revealing a low variation in the oxide thickness for both samples, varying from 1.6 to 3.0 nm for the AZ sample, which increased from 1.2 to 5.4 nm for the AZ + Ar-P sample. The thickness changes slightly with the immersion time, revealing the formation and detachment of this layer.

Leleu et al. [31] reported that exposure of pure magnesium and a WE34 magnesium alloy to a 0.1 M Na_2_SO_4_ solution promotes, after time, an increase in the thickness of the protective layer. The authors also reported a decrease in the thickness of this layer after 72 h of immersion, which was attributed to the rupture and detachment of the outer MgO layer [2]. Considering the above, the high initial layer thickness on the AZ + Ar-P surface may be related to the increased MgO coverage, as described in the XPS analysis. Therefore, argon plasma increases the activity on the surface.

Figure 9 shows the equivalent circuit and physical model proposed for a better understanding of the corrosion process of AZ31 alloy immersed in 0.1 M of Na_2_SO_4_ solution, showing the influence of the plasma argon pretreatment. In this case, R_e_ is the resistance of the electrolyte between the working and the reference electrode, R_CT_ represents the charge transference resistance, CPE_CT_ is the double layer and MgO barrier film capacitance, R_film_ and CPE_film_ represent the diffusion of Mg^2+^ and oxygen through the outer film (Mg(OH)_2_), L represents the inductance, and R_L_ is the inductance resistance caused by the intermediate species on the surface of the alloy without a corrosion layer, as explained above.

Table 4 shows the evolution of the fit parameters using the equivalent circuit proposed in Figure 9, revealing that the argon plasma has a small effect on the R_CT_, which increases in shorter times, as reported by Ascencio et al. [23] and Leleu et al. [2] for WE43 alloy. The authors proposed that this increment can be associated with the film formation on the surface and its protective properties, generally related to MgO. The reduction of the R_CT_ suggests the rupture of the film and the increase of the free-film area. The maximum R_CT_ for AZ samples was around 16 kΩ cm^2^ after 120 h, which decreased to 8.5 kΩ cm^2^. Meanwhile, for AZ + Ar-P samples, the R_CT_ decreased until 10 kΩ cm^2^ after 24 h, which can be explained by the initial reactivity of the surface, gradually increasing to a value near 14 kΩ cm^2^, which was associated with the formation of the film, which could be more stable than that formed without pretreatment for longer times.

Table 4 shows that the alpha value related to the double layer capacitance was close to 0.93 for both samples, which can be attributed to their heterogeneous surfaces, which was not influenced by the immersion time. Furthermore, the CPE coefficient showed a variation around 2 × 10^−5^ Fs^(α−1)^cm^−2^ for AZ samples, and for AZ + Ar-P samples the variation is between 1 to 5 × 10^−5^ Fs^(α−1)^cm^−2^. This coefficient is attributed to the deviation of the ideal capacitor; due to heterogeneities of the surface, these heterogeneities can be a high roughness, distributed surface reactivity, electrode porosity, and current and potential distributions associated with electrode geometry [39,40]. Additionally, there is a low variation of the R_film_, revealing the stability of the oxide film over the immersion time for both samples, suggesting a protective layer, associated with MgO [29,31,41] and/or Mg(OH)_2_ [23,29,31]. Notice that the AZ samples had a higher dispersion in the R_film_, as described in the polarization curves section.

### 3.3. Weight Loss

Figure 10 shows the weight loss of AZ31 alloy in Na_2_SO_4_ solution, revealing that after 3 h of immersion, the corrosion rate decreased when the pretreatment was applied, while for the longer exposure time, there was no difference on both surfaces, showing a decrease in the corrosion rate is observed over short times with a tendency to stabilize after 72 h. The decrease in the corrosion rate can be attributed to the formation of the partially protective oxide layer [42], in agreement with electrochemical measurements. The mass loss increases with time, suggesting that corrosion kinetics for both surfaces are maintained, despite the formation of a surface film, according to Pardo and Tian [43,44]. Xin et al. [38], studying the weight loss of AZ91 magnesium alloy in SBF for different exposure times, found a behavior of the corrosion rate similar to that found employing impedance measurements. Thus, at the beginning of the experiment, the authors reported a rapid decrease in the corrosion rate, which stabilized over time when equilibrium was established between the formation and dissolution of the surface layer. Tian et al. [44] studied the AZ91 magnesium alloy in sodium sulfate solutions at different pH values, from pH 2 to 12. The alloy showed the same behavior as that reported by Xin et al. [38]; the difference between the corrosion rate in different pH solutions was for shorter exposition times, revealing a higher corrosion rate at lower pH values.

In summary, surface analysis shows that argon plasma pretreatment increases the amount of oxygenated species over hydroxylated ones. The results obtained with the electrochemical measurements confirm a greater coverage of MgO on the alloy surface, which is reflected in a more significant protective character of the oxide film that covers the surface of the alloy. The increase in initial reactivity of the surface promotes an increase in the thickness of the surface film in the first hours of exposure to the electrolyte, decreasing the corrosion rate, verified by mass loss measurements. This can be seen schematically in Figure 11.

## 4. Conclusions

The main effect of the argon plasma pretreatment on AZ31 alloy is the increase in the wettability and surface energy. The ion collision modified surface topography that is characterized by increased surface roughness. The chemical change is given by the increase of the O/OH^−^ ratio i.e., a higher coverage of the MgO film on the surface after pretreatment, due to the initial increase of activity of the surface. In summary, there is a rougher surface, with greater MgO coverage and with more active sites.

The electrochemical measurements show that the argon plasma pretreatment does not influence the corrosion mechanism, at least for a short exposure time, as was revealed in OCP measurements. However, polarization curves showed an inflection point associated with the formation and rupture of the surface film that shifted to more positive potential values. The increase in the potential was more accentuated after the pretreatment; therefore, the pretreatment induces the formation of a passive oxide film, as revealed by XPS analysis. EIS measurements showed that the initial higher thickness of the AZ + Ar-P surface layer might be related to the increase in MgO coverage due to the increase of the initial reactivity, represented by the equivalent circuit analysis.

Weight loss measurements initially showed a decrease in the corrosion rate after pretreatment, likely related to the increase of the activity and the increase of the initial film thickness.

## Figures and Tables

**Figure 1 materials-16-02327-f001:**
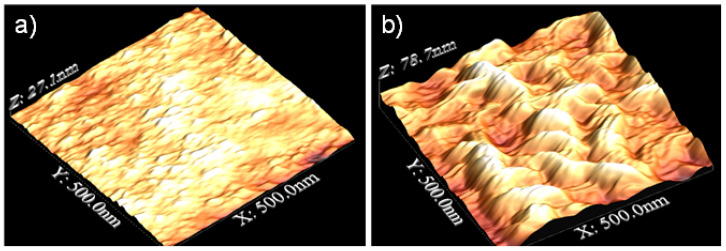
AFM images of AZ31 alloy (**a**) before and (**b**) after 2 min of pretreatment.

**Figure 2 materials-16-02327-f002:**
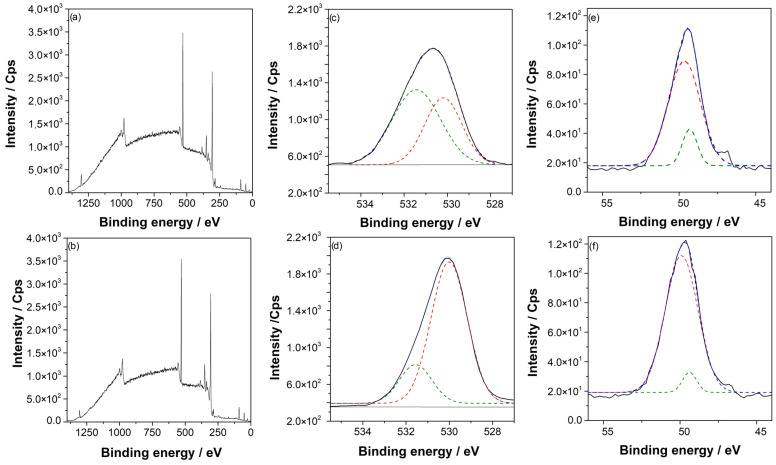
(**a**) Survey spectra for AZ samples, (**b**) survey spectra for AZ + Ar-P samples, (**c**) high-resolution XPS of O 1s spectra for AZ samples, (**d**) high-resolution XPS of O 1s spectra for AZ + Ar-P samples, (**e**) high-resolution XPS of Mg 2p spectra for AZ samples, and (**f**) high-resolution XPS of Mg 2p spectra for AZ + Ar-P samples.

**Figure 3 materials-16-02327-f003:**
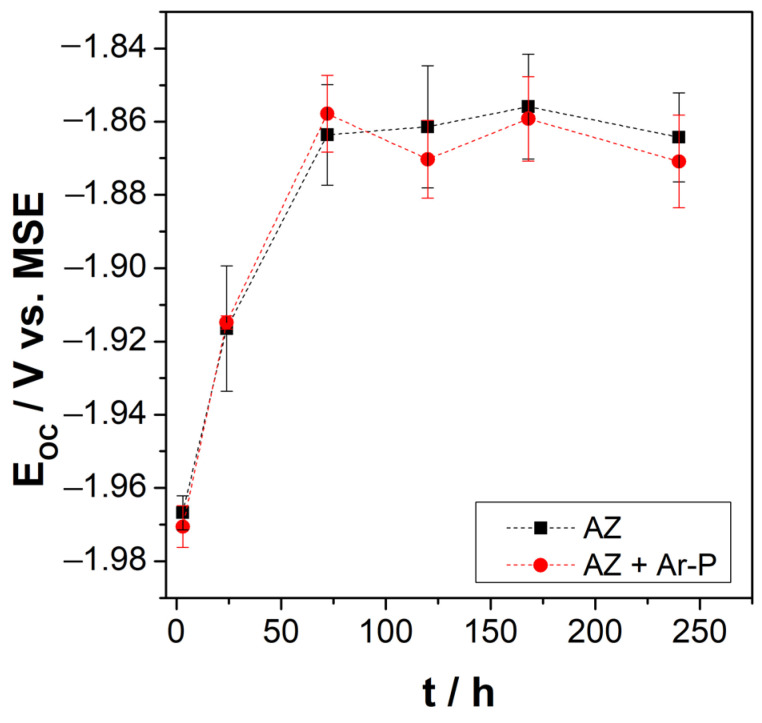
Open circuit potential of AZ31 surfaces (before and after pretreatment) as a function of immersion time in Na_2_SO_4_ 0.1 M. (■) AZ and (●) AZ + Ar-P samples.

**Figure 4 materials-16-02327-f004:**
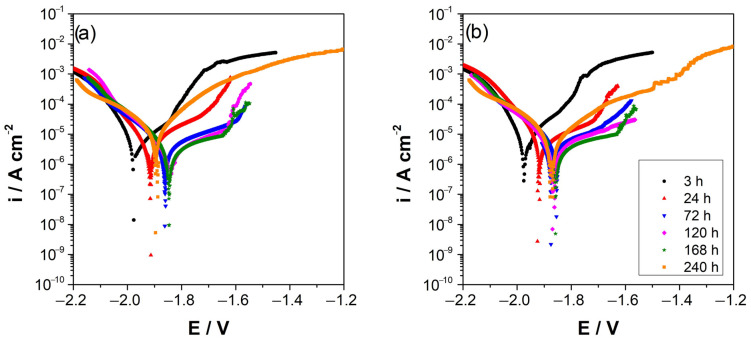
Potentiodynamic polarization curves of AZ31 alloy immersed in 0.1 M Na_2_SO_4_, (**a**) AZ surface, (**b**) AZ + Ar-P surface. All potentials refer to MSE electrode.

**Figure 5 materials-16-02327-f005:**
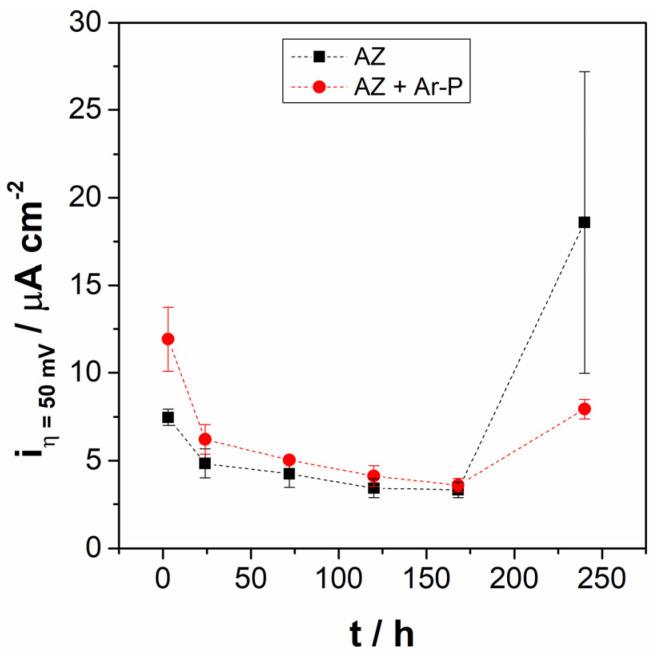
Evolution of anodic current density at +50 mV vs. E_OC_, in the pseudo-passive range for AZ and AZ + Ar-P samples.

**Figure 6 materials-16-02327-f006:**
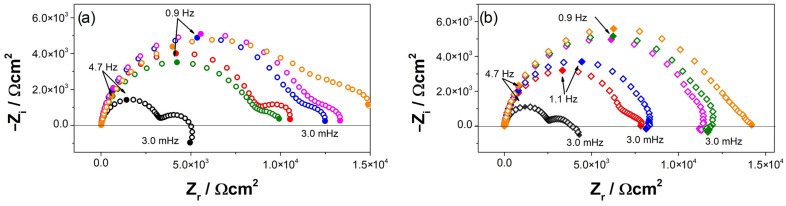
Nyquist diagrams evolution of (**a**) AZ and (**b**) AZ + Ar-P samples in 0.1 M of Na_2_SO_4_, (○,◊) 3 h, (○,◊) 24 h, (○,◊) 72 h, (○,◊) 120 h, (○,◊) 168 h, and (○,◊) 240 h.

**Figure 7 materials-16-02327-f007:**
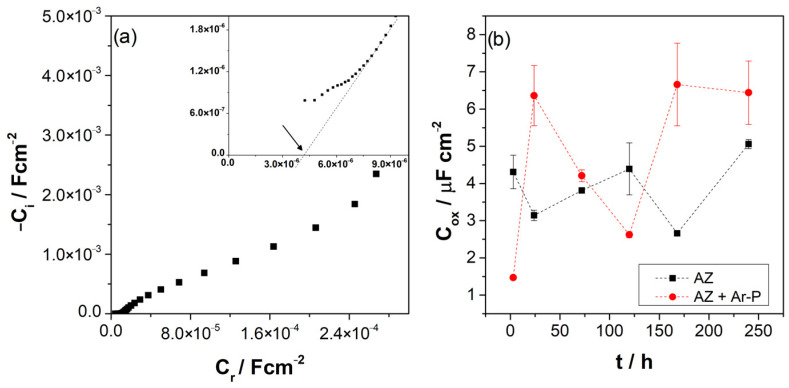
(**a**) A representative Cole-Cole plot of AZ samples after 3 h of immersion, as an example for the determination of C_ox_; (**b**) Evolution of complex capacitance (C_ox_) for the differently pretreated AZ31 alloy immersed in 0.1 M of Na_2_SO_4_.

**Figure 8 materials-16-02327-f008:**
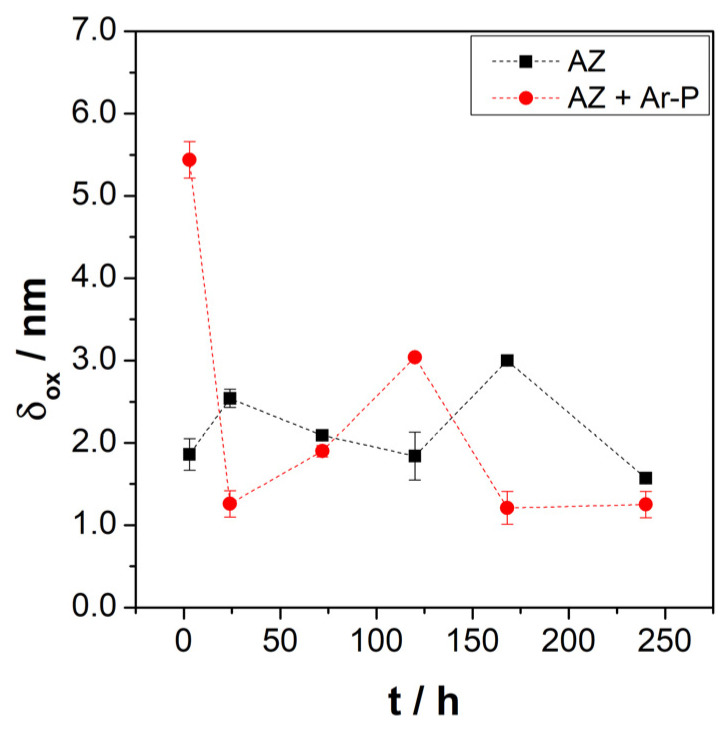
Evolution of the layer thickness (δ_ox_) for the differently pretreated AZ31 alloy immersed in 0.1 M of Na_2_SO_4_.

**Figure 9 materials-16-02327-f009:**
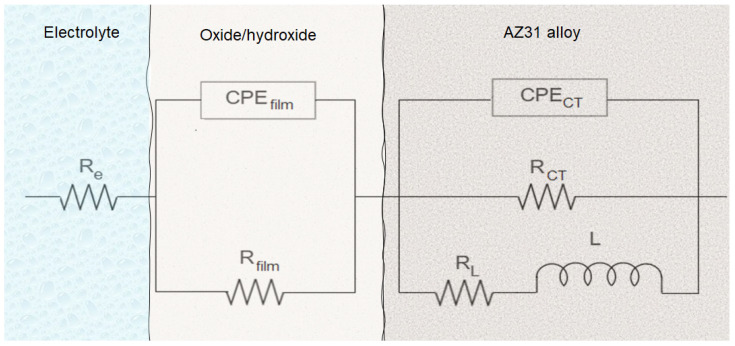
Proposed equivalent circuit and physical model representing the impedance responses of samples AZ and AZ + Ar-P after exposure in 0.1 M Na_2_SO_4_ solution.

**Figure 10 materials-16-02327-f010:**
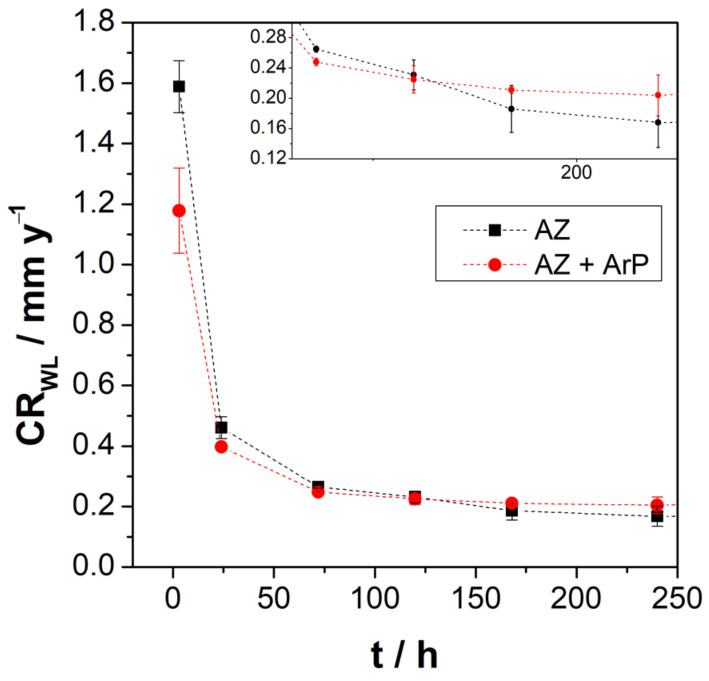
Corrosion rate vs. immersion time determined by weight loss experiments.

**Figure 11 materials-16-02327-f011:**
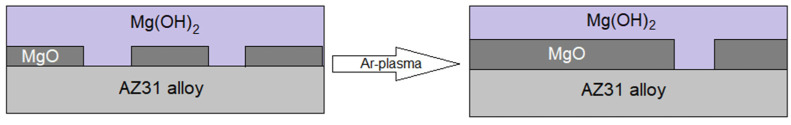
Schematic illustration of the cross-section view of the oxide/hydroxide film formed on the AZ31 alloy surfaces, before and after the pretreatment.

**Table 1 materials-16-02327-t001:** Contact angle (θ) and surface energy (γ), before and after argon plasma pretreatment.

t s	θ_w_ °	θ_D_ °	γ_LV_ mN m^−1^	γ^d^_LV_ mN m^−1^	γ^p^_LV_ mN m^−1^
0	37.60 ± 8.37	42.67 ± 7.42	64.12 ± 2.80	38.08 ± 3.90	26.04 ± 6.70
120	5.31 ± 0.98	11.76 ± 2.90	80.01 ± 0.17	49.71 ± 0.52	30.94 ± 0.35

**Table 2 materials-16-02327-t002:** Electrochemical parameters of AZ31 alloy in Na_2_SO_4_ 0.1 M.

Sample	t h	E_corr_ V_MSE_	i_corr_ μA cm^−2^	E_B_ V_MSE_	∆E_pass_ V
AZ	3	−1.96 ± 0.01	3.8 ± 0.1	−1.82 ± 0.03	0.11 ± 0.01
	24	−1.93 ± 0.01	2.9 ± 0.4	−1.72 ± 0.02	0.16 ± 0.02
	72	−1.87 ± 0.02	3.2 ± 0.7	−1.66 ± 0.05	0.21 ± 0.06
	120	−1.83 ± 0.02	2.9 ± 0.5	−1.63 ± 0.08	0.17 ± 0.06
	168	−1.82 ± 0.02	3.6 ± 0.5	−1.63 ± 0.03	0.16 ± 0.09
	240	−1.93 ± 0.06	21.8 ± 10.4	-	0
AZ + Ar-P	3	−1.97 ± 0.01	6.6 ± 2.6	−1.83 ± 0.01	0.09 ± 0.02
	24	−1.91 ± 0.01	5.9 ± 0.03	−1.71 ± 0.01	0.17 ± 0.01
	72	−1.85 ± 0.01	4.3 ± 0.6	−1.68 ± 0.01	0.13 ± 0.01
	120	−1.85 ± 0.02	3.9 ± 0.6	−1.69 ± 0.02	0.21 ± 0.09
	168	−1.82 ± 0.04	4.5 ± 0.3	−1.61 ± 0.03	0.22 ± 0.03
	240	−1.90 ± 0.03	11.0 ± 0.3	−1.46 ± 0.06	0.28 ± 0.01

**Table 3 materials-16-02327-t003:** Impedance parameter obtained graphically for AZ31 alloy in Na_2_SO_4_ 0.1 M.

Sample	t h	R_e_ Ωcm^2^	−α_HF_	Q_eff_ × 10^−5^ Fs^(α−1)^cm^−2^	C_ox_ μFcm^−2^
AZ	3	63.0 ± 2.6	0.90 ± 0.01	1.70 ± 0.22	4.31 ± 0.45
	24	69.2 ± 3.7	0.91± 0.01	2.25 ± 0.02	3.14 ± 0.14
	72	73.4 ± 0.8	0.88 ± 0.02	2.77 ± 0.52	3.81 ± 0.03
	120	79.8 ± 6.6	0.88 ± 0.02	2.52 ± 0.49	4.39 ± 0.70
	168	79.3 ± 4.8	0.86 ± 0.02	2.58 ± 0.71	2.66 ± 0.01
	240	81.1 ± 9.2	0.86 ± 0.03	2.63 ± 0.60	5.06 ± 0.12
AZ + Ar-P	3	66.0 ± 1.8	0.90 ± 0.03	1.74 ± 0.47	1.47 ± 0.06
	24	76.0 ± 4.9	0.90 ± 0.02	2.41 ± 0.24	6.36 ± 0.81
	72	73.0 ± 6.1	0.90 ± 0.03	2.20 ± 0.12	4.21 ± 0.16
	120	81.8 ± 4.6	0.84 ± 0.02	2.78 ± 0.49	2.62 ± 0.08
	168	74.3 ± 7.2	0.87 ± 0.03	2.44 ± 0.17	6.66 ± 1.11
	240	76.2 ± 8.3	0.89 ± 0.02	2.26 ± 0.15	6.44 ± 0.85

**Table 4 materials-16-02327-t004:** Impedance fit parameters obtained AZ31 alloy in 0.1 M Na_2_SO_4_ solution.

Sample	t h	R_e_ Ωcm^2^	R_CT_ kΩcm^2^	−α_CT_	CPE_CT_ x 10^−5^ Fs^(α−1)^cm^−2^	C_CT_ μFcm^−2^	R_film_ kΩcm^2^	−α_film_	CPE_film_ mFs^(α−1)^cm^−2^	C_film_ mFcm^−2^
AZ	3	63.0 ± 2.6	2.4 ± 0.3	0.91 ± 0.01	1.4 ± 0.6	4.5 ± 0.1	3.3 ± 0.4	0.77 ± 0.04	1.6 ± 0.4	2.9 ± 0.7
	24	69.2 ± 3.7	7.8 ± 1.1	0.94 ± 0.03	1.8 ± 0.6	17.8 ± 0.5	1.1 ± 0.1	0.88 ± 0.15	5.9 ± 2.1	7.9 ± 0.8
	72	73.4 ± 0.8	10.1 ± 0.7	0.92 ± 0.01	2.0 ± 0.1	5.6 ± 2.8	1.3 ± 0.4	0.99 ± 0.01	8.4 ± 3.3	8.8 ± 3.8
	120	79.8 ± 6.6	15.9 ± 0.4	0.93 ± 0.01	1.9 ± 0.6	21.4 ± 5.1	2.3 ± 1.4	0.66 ± 0.09	0.9 ± 0.1	1.7 ± 0.7
	168	79.3 ± 4.8	8.9 ± 1.0	0.93 ± 0.02	0.2 ± 0.1	1.4 ± 0.6	1.3 ± 0.9	0.69 ± 0.19	0.7 ± 0.1	1.6 ± 0.9
	240	81.1 ± 9.2	8.5 ± 2.1	0.91 ± 0.01	3.2 ± 1.6	28.3 ± 15.2	3.6 ± 3.4	0.81 ± 0.13	0.9 ± 0.2	1.1 ± 0.2
AZ + Ar-P	3	66.0 ± 1.8	2.6 ± 0.4	0.92 ± 0.01	0.9 ± 0.3	7.8 ± 1.3	3.0 ± 1.0	0.73 ± 0.22	1.8 ± 0.8	3.3 ± 0.9
	24	76.0 ± 4.9	9.8 ± 0.8	0.95 ± 0.02	1.7 ± 0.6	18.2 ± 0.1	1.6 ± 0.8	0.91 ± 0.09	2.7 ± 1.3	5.6 ± 1.9
	72	73.0 ± 6.1	8.3 ± 0.4	0.94 ± 0.01	2.0 ± 0.2	16.7 ± 0.5	2.4 ± 0.7	0.89 ± 0.03	4.0 ± 0.8	4.1 ± 3.7
	120	81.8 ± 4.6	9.4 ± 2.2	0.91 ± 0.02	5.8 ± 3.1	4.4 ± 2.3	1.0 ± 0.1	0.60 ± 0.23	0.8 ± 0.1	0.6 ± 0.1
	168	74.3 ± 7.2	11.2 ± 0.6	0.93 ± 0.01	1.2 ± 0.7	6.7 ± 1.4	1.1 ± 0.1	0.79 ± 0.21	4.7 ± 1.7	0.4 ± 0.1
	240	76.2 ± 8.3	13.6 ± 1.3	0.94 ± 0.02	2.6 ± 0.8	20.0 ± 2.7	2.3 ± 0.1	0.63 ± 0.18	3.3 ± 1.2	-

## Data Availability

The raw/processed data required to reproduce these findings cannot be shared at this time due to technical or time limitations.

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
