# Peer review of "Effect of Plasma Argon Pretreatment on the Surface Properties of AZ31 Magnesium Alloy"

_materials, 2023, doi:10.3390/ma16062327_

Round 1
Reviewer 1 Report
the present work investigated the influence of argon plasma pretreatment on the surface properties of an AZ31 magnesium alloy focus on the enhancement of the reactivity of the surface, using surface analysis techniques, electrochemical techniques, and gravimetric measurements. Contact angle measurements revealed higher surface energy after applying the pretreatment, and atomic force microscopy showed a roughness increase, while X-Ray photoelectron spectroscopy showed the chemical change on the surface, where after pretreatment increase the oxygen species. Electrochemical impedance spectroscopy showed that the pretreatment does not affect the corrosion mechanism of the alloy but it can influence the formation of oxide layer. Weight loss measurement exhibited a rapid decrease of the corrosion rate in early stage and a stabilized the corrosion rate after 72 h, which was associated with the formation of a film on the alloy.
I had carefully revised the manuscript and my decision is to be revised with minor revisions.
Comments:
1- Abstract part needs to be rewritten to include identically 3 brief informative subheadings, Background, Methods, and Significant Findings from the applied characteristic methods.
2- SEM images for AZ31 magnesium alloy surface before and after argon plasma pretreatment.
3- An expected mechanism displayed the effect of plasma argon pretreatment on the surface properties of AZ31 magnesium alloy.
Reviewer 2 Report
The manuscript “Effect of plasma argon pretreatment on the surface proper-2 ties of AZ31 magnesium alloy” by Cecilia Montero et al. concerns important subject of corrosion resistance of magnesium alloys. The manuscript is interesting and well prepared. Nevertheless, I would like to propose a few corrections before publishing.
The issue which I consider important is better description of what is the novelty of the results presented in the manuscript. There is one statement “In the case of magnesium alloys, the study is very novel” however in view of the discussion in which nearly all reported properties are explained using the literature data this seems not sufficient.
Page 9: What do the Authors mean by “intermetallic particles”?
Page 11: In the sentence “Considering the above, the initial increase in the thickness of the AZ + Ar-P samples may be related to the increase in MgO coverage, as described in the XPS analysis” it should be rather written “the initial higher thickness of the layer on the AZ+Ar-P sample surface may be related…”
Fig. 9: hydroxide not hidroxide.
Page 12: The sentence “Table 4 shows the evolution of the fit parameters using the equivalent circuit proposed in Figure 9, revealing that the argon plasma has not a significant effect on the RCT, which increases in shorter times, as reported by Ascencio et al. [21]..” , is not clear. Below this sentence the Authors describe the differences between the variations in Rct of both samples.
Page 13: I believe, the last sentence, i.e. “However, the pretreatment induces the formation of a passive oxide film, in accordance with XPS analysis” is the main conclusion of the work. Why is it so weakly stressed (not stressed at all)?
Reviewer 3 Report
1. On what criteria the power has been decided to the electrode 80W should be given in the experimental section, even though it was cited.
2. What are the main reasons for varying in surface roughness?
3. Please provide the surface microstructure after pretreatment and before, in order to distinguish the roughness surfaces by AFM.
4. As per Fig 3, there is no much difference in the ECor between Az and Az+Pre. Justification needed?
5. Conclusions should be arranged in an order and provide the key results of all the results.
6. Introduction should be improved by adding some more recent relevant studies.
Author Response
- On what criteria the power has been decided to the electrode 80W should be given in the experimental section, even though it was cited.
Answer: Considering that the objective of the work is to activate the surface and not to etch it, a previous study was carried out in which it was determined that 80 W was a sufficient power to activate the surface without etching it and that the half-life of the radicals was the long enough to be able to react on the surface. Clarifying text added on page 3, lines 112-114:
In order to activate the surface, argon plasma pretreatment was performed on half of samples with a Plasma Prep III device (SPI, Plasma Prep III Solid State, West Chester, USA) that consisted of an RF supply (13.56 MHz) in a dielectric quartz tube in a hollow electrode. The gas flowing out of the nozzle formed a plasma jet of pure argon (5 L/min). The power applied to the electrode was adjusted to 80 W at a pressure of around 150 mTorr (20 Pa). This power was chosen because it is the one that provides enough power to activate the surface without etching it. The exposition time was two minutes, according with the protocol proposed by Muñoz et al. [7].
- What are the main reasons for varying surface roughness?
Answer: Increase the contact area to improve the adhesion of anticorrosive coatings
- Please provide the surface microstructure after pretreatment and before, in order to distinguish the roughness surfaces by AFM.
Answer: Because the objective of the work is to increase the activity of the surface and not to carry out pickling, the roughness changes are of the order of nanometers (from 4 nm before pretreatment to 16 nm after pretreatment), so the changes cannot be seen under a microscope, which is why AFM surface analysis was chosen.
- As per Fig 3, there is no much difference in the ECor between Az and Az+Pre. Justification needed?
Answer: We appreciate your observation. Considering that the changes in the surface roughness are of the order of nanometers, it is reasonable that such changes do not generate changes in the OCP, especially considering that the potential value will be an average over the entire area and the entire exposure time. The differences could be evidenced at overpotentials but not at rest. Considering his observation, a text was added to the manuscript (from page 6, line 251, to page 7 line 275)
The open circuit potential (OCP) measurements of the AZ and AZ + Ar-P samples were performed after different exposure times to the electrolyte, up to a maximum exposure time of 240 h. The behavior of the OCP over time is presented in Figure 3. For exposure times of less than 48 h, the OCP moves progressively towards more positive values; for over 48 h of exposure to the electrolyte, the OCP reaches a constant value close to -1.86 V vs. SSE. This behavior of the OCP over time agrees with that reported by Leleu et al. [2]. OCP stabilization has been commonly attributed to developing a partially protective film on the metal surface composed of MgO/Mg(OH)2 [2,27–29]. The behavior of the pH with the exposure time at the surface-electrolyte interface also coincides with that reported by Leleu et al. [2]; the pH increases from 7 to ~10 after 24 h of immersion. The increase in pH may be related to the formation and dissolution of the Mg(OH)2 film and, consequently, to the release of OH- to the electrolyte, as described by Baril et al. [30]. According to previous studies, the maximum potential reached corresponds to the stabilization of the hydroxylated species and is related to a pH value of around 10.5 [31]. The OCP behaviors of the AZ and AZ + Ar-P samples are similar, which is related to the magnitude of the impact of the plasma pretreatment on the surface roughness. As revealed by the AFM micrographs in the Figure, the changes in roughness promoted by the plasma pretreatment are nanometers.
In this investigation, argon plasma pretreatment was used only to activate the surface, and changes in roughness at the nanometer scale would not affect the OCP values. There needs to be more than the differences in roughness to resolve the small changes that can affect the system. For this reason, other electrochemical techniques are used to determine changes in the electrochemical behavior of samples.
- Conclusions should be arranged in an order and provide the key results of all the results.
Answer: We highly appreciate your observation; the conclusion was modified as follows (Page 14 to 15):
The main effect of the argon plasma pretreatment on AZ31 alloy is the increase in the wettability and surface energy. The ion collision modified surface topography that is characterized by increased surface roughness. The chemical change is given by the increase of the O/OH- ratio i.e., a higher coverage of the MgO film on the surface after pretreatment, due the initial increase of activity of the surface. In summary, there is a rougher surface, with greater MgO coverage and with more active sites.
The electrochemical measurements show that the argon plasma pretreatment does not influence the corrosion mechanism, at least for a short exposure time, as was revealed in OCP measurements. However, polarization curves showed an inflection point associated with the formation and rupture of the surface film that shifted to more positive potential values. The increase in the potential was more accentuated after the pretreatment; therefore, the pretreatment induces the formation of a passive oxide film, as revealed by XPS analysis. EIS measurements showed that the initial higher thickness of the AZ + Ar-P surface layer might be related to the increase in MgO coverage due to the increase of the initial reactivity, represented by the equivalent circuit analysis.
Weight loss measurements initially showed a decrease in the corrosion rate after pre-treatment, likely related to the increase of the activity and the increase of the initial film thickness.
- Introduction should be improved by adding some more recent relevant studies.
Answer: Thank you very much for your appreciation. It was challenging to find new works due to the novelty of the subject, but we found two recent studies from 2020 and 2022, which were added on page 2, from lines 85 to 89.
Xu et al. [14] report an improvement in polyphenylene adhesion, and an augment in the fracture resistance of the composite structure. On the other hand, Yoshida et al. [15] used oxygen plasma to improve the adhesion of the polyether ether ketone film to the copper surface, increasing hydrophilicity and surface roughness.
In Reference (page 15, line 548-553)
- Xu, D.; Yang, W.; Li, X.; Hu, Z.; Li, M.; Wang, L. Surface Nanostructure and Wettability Inducing High Bonding Strength of Polyphenylene Sulfide-Aluminum Composite Structure. Appl Surf Sci 2020, 515, 145996, doi:10.1016/J.APSUSC.2020.145996.
- Yoshida, M.; Nakanishi, G.; Yamanaka, H.; Iwamori, S. Enhanced Adhesion of Copper Plating to Polyether Ether Ketone Based on Active Oxygen Species Generated under Ultraviolet Irradiation. Surface and Interface Analysis 2022, 54, 759–766, doi:10.1002/SIA.7088.
Round 2
Reviewer 3 Report
Revised version is improved in its quality and addressed all of my concerns.